# Application of Deep Learning in the Deployment of an Industrial SCARA Machine for Real-Time Object Detection

**Tibor Péter Kapusi** [1] , **Timotei István Erdei** [2,*], **Géza Husi** [2] and **András Hajdu** [1]

1   Faculty of Informatics, Department of Data Science and Visualization, University of Debrecen, Kassai Str. 26, 4028 Debrecen, Hungary; kapusi.tibor@inf.unideb.hu (T.P.K.); hajdu.andras@inf.unideb.hu (A.H.)
2   Faculty of Engineering, Department of Air- & Road Vehicles, University of Debrecen, Ótemető Str. 2–4, 4028 Debrecen, Hungary; husigeza@eng.unideb.hu
*   Correspondence: timoteierdei@eng.unideb.hu; Tel.: +36-52-415-155

**Abstract:** In the spirit of innovation, the development of an intelligent robot system incorporating the basic principles of Industry 4.0 was one of the objectives of this study. With this aim, an experimental application of an industrial robot unit in its own isolated environment was carried out using neural networks. In this paper, we describe one possible application of deep learning in an Industry 4.0 environment for robotic units. The image datasets required for learning were generated using data synthesis. There are significant benefits to the incorporation of this technology, as old machines can be smartened and made more efficient without additional costs. As an area of application, we present the preparation of a robot unit which at the time it was originally produced and commissioned was not capable of using machine learning technology for object-detection purposes. The results for different scenarios are presented and an overview of similar research topics on neural networks is provided. A method for synthetizing datasets of any size is described in detail. Specifically, the working domain of a given robot unit, a possible solution to compatibility issues and the learning of neural networks from 3D CAD models with rendered images will be discussed.

**Keywords:** cyber-physical systems; Industry 4.0; SCARA robot; deep learning; YOLO

## 1. Introduction

The COVID-19 pandemic and the global chip shortage has created the need for old industrial machine units on production lines to use some form of deep learning, as is used in Industry 4.0.

We have a large number of modern and old machine units, and part of our research aims to extend the applicability of these devices. Generally speaking, older robot models can be considered obsolete in the industry in many ways. Most of them were not designed to meet modern standards at the time of commissioning, nor did their controller allow them to communicate/be controlled via a network [1].

Transforming obsolete robots is a challenge in itself, as most of the time the underlying operating system is embedded and the hardware specification does not allow for the performance of sophisticated image-analysis tasks.

With this in mind, a principle has been developed that allows the operation of old machine units supplemented with deep learning without additional financial outlay, thus setting the application of these units on a new footing. In our case, the robot unit, the function of which is to be supplemented by deep learning, is an SRX-611 manufactured by SONY in Tokyo, Japan (see Figure 1), previously widely used.

Robots are widely used in manufacturing and can be categorized according to several aspects; their design, work area, control and auxiliary energy source for operation all act as influencing factors on the applicability of a given robot.

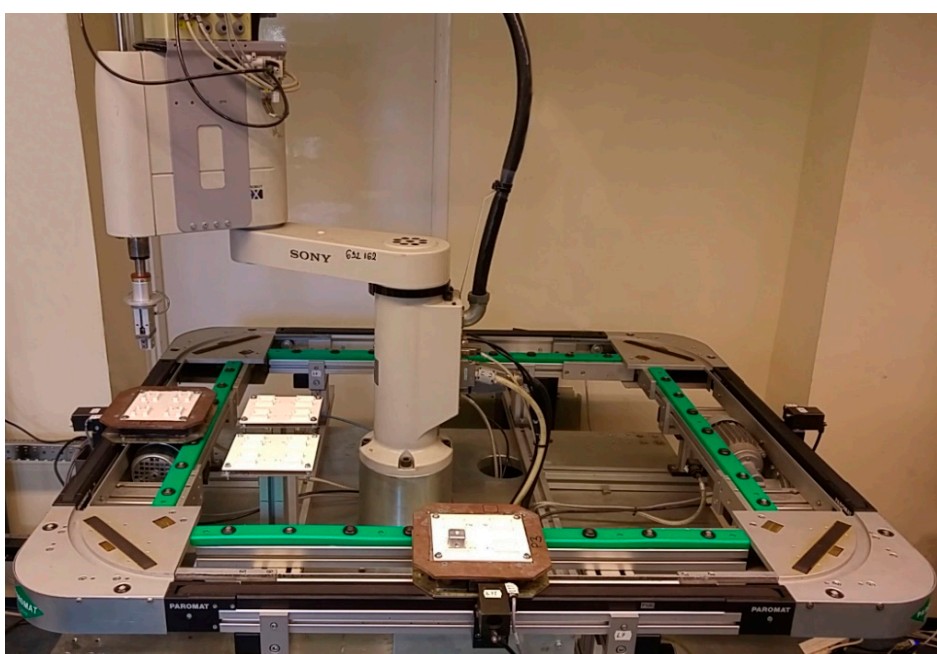

**Figure 1.** Sony SRX-611 CPS-LAB robotic arm for smartening.

It can be stated that the main reasons for using industrial robot units re economic, as detailed in the annual report of the International Federation of Robotics (IFR) [2]:

- Predictability;
- Increase in productivity;
- Flexible production;
- Decrease in scrap products;
- Lower operating costs;
- Better product quality.

The above do not follow as a rule from the integration of an industrial robot into a production line, but they represent the current aspirations of industrial production and together determine the direction of its development.

In industrial systems, it is important to consider the worst-case scenarios that can occur and cause accidents during operations. Several relevant studies have been conducted using recurrent neural network-based approaches to investigate this.

Several machine learning-based methods for avoiding collisions between robot manipulators have been developed by several other authors' research. The motion coordination algorithm is handled by a recurrent network, whose convergence is a requirement for solving the constrained quadratic optimization problem, which allows the redundant manipulator systems to be expressed, whether SCARA robots or robotic arms.

In our case, it is worth mentioning that the deep learning technique used is not specifically linked directly to the Sony SCARA controller. As a result, the user will be informed about the presence of objects in the robot's workspace based on later results of object detection.

The robot does not perform an obstacle-avoidance maneuver depending on the outcome. In the case of trajectory planning, several approaches are used to avoid obstacles. For such a purpose, the RNN (also known as the recurrent neural network approach) is used, capable of both target tracking and collision avoidance simultaneously. Since it permits time-dynamic behavior, multi-axis robot arms have been successfully used for trajectory planning and obstacle avoidance.

In another research study, a recurrent neural network for collision avoidance was used to define a given robot unit and the obstacles in its workspace as a set of critical points. The obstacle avoidance itself can be described by class-K functions, but the QP (quadratic

programming) problem must be handled at the speed level. In this case, the RRN neural network is trained to handle QP. However, its pretesting was performed in a simulation environment and the parameters of a virtual plane 4-DOF robot arm were derived. Since we are dealing with planar motion here, the obstacle to be avoided was also defined as a single point [3].

The fact that neural networks can be applied not only to static objects but also to moving objects demonstrates their versatility. When a problem is not only modeled virtually but also translated into physical form, it takes on a completely different complexion. Furthermore, if we are not dealing with a fixed robot arm, but with a moving object or an autonomous vehicle, the problem must be approached using entirely different principles, although the two applications have similar starting points, such as the problem of path assignment and tracking defined as quadratic programming. That is, the redundant resolution problem is viewed as a quadratic programming minimization scheme. Thus, several goals can be achieved more efficiently, such as target tracking and repetitive motion planning. However, in the case of moving objects, especially if the main unit is also moving, it is necessary to use a vision sensor for the detection. An ANN (artificial neural network) can be used in such cases to avoid the objects. Such systems are also known as universal function approximators because they are suitable for nonlinear function training [4].

It is important to note that some studies have shown that a robot unit can also use vision sensors to implement obstacle avoidance. CCD camera images placed in the robot workspace can be used to detect standard fixed objects. The physical robot unit used for a test was a 7-DOF KUKA LBR iwa. The real-time motion control of a physical robot unit is a complex problem. Since, in this situation, we are not dealing with a 2D dimensional case, the limitation of the axes and the predefined trajectory of the end-effector must be considered. Furthermore, the robot has singularity points that affect all axes except the A7, implying the loss of degrees of freedom. RNN was used to develop the motion-planning method. The robot's workspace was downscaled, and the angular and joint velocity limits were specified [5].

In addition to the above, there are also potential applications of neural networks at the level of collaborative SCARA robots. There are a variety of industrial applications for specialized robot units in which other people perform tasks in their workspaces or where multiple robots perform coordinated tasks in each other's workspaces. The objective, once again, is to design a path that has no chance of resulting in collisions, even at high speeds. This is a kinematic control problem with multiple factors to consider. Quadratic programming is the starting point in this case, but RNN acts as a dynamic controller. The principle is based on the fact that when two SCARA robots are moving along a given trajectory and the preset distance between them has decreased at a point along the trajectory, they begin to swerve to avoid colliding [6]. The preceding examples demonstrate the numerous applications of neural networks in robotics.

The rest of the paper is organized as follows. Section 2 details the construction and operational information for an older robot unit, the SONY SCARA-SRX-611. Information on the programming language of the robot unit and its currently supported interfaces is explained in more detail in Section 3. Section 4 introduces the procedures for implementing modeling and the software platforms used. Details of the synthesis of image datasets and a description of the algorithms used for their generation and the rendering process itself will be provided in Section 5. The chosen neural network architecture and the detector we designed, Robonet-RT, are detailed in Section 6. Section 7 describes the process of training the detector, and, finally, Section 8 presents the results obtained during the learning process, together with the selected metrics and the loss function. Conclusions are drawn in Section 9.

## 2. SONY SCARA SRX-611 Robot Unit

The SCARA unit is based on Hiroshi Makino's 1978 concept and the name has a complex meaning: Selective Compliance Assembly Robot Arm [7,8].

SCARA robots are designed to operate at higher axis speeds and allow more precise work. One of the defining parameters of machines is the number of axes and their type. The SONY SCARA has three axes, two rotary and one translational.

The RRT design also defines the robot's workspace, where the robot can perform assembly/implantation tasks. All the parameters are given precisely in Figure 2 and Table 1.

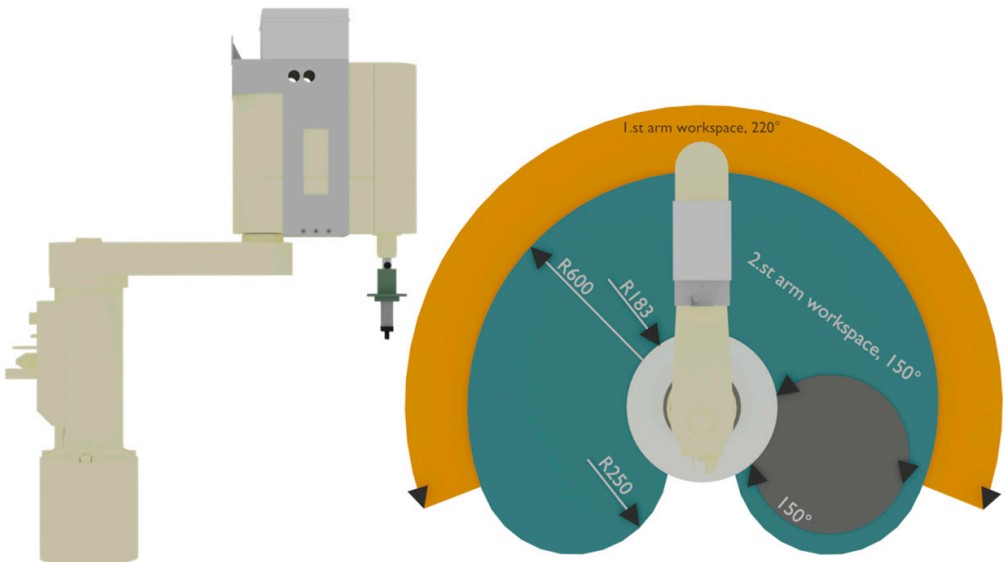

**Figure 2.** SONY SCARA SRX-611 workspace and axes parameters.

**Table 1.** SONY SCARA SRX parameters [8].

| Arm Length | | Workspace | | Cycle Time | Maximum Speed | | Repeatability of Position | |
|---|---|---|---|---|---|---|---|---|
| 1. Axis | 350 mm | 1. Axis | 220° | 0.6 [s] | | | | |
| 2. Axis | 250 mm | 2. Axis | ±150° | | | | | |
| | | Z. Axis | 150 mm | | Z axis | 770 mm/s | Z axis | ±0.02 mm |
| | | R. Axis | ±360° | | R axis | 1150/s | R axis | ±0.03 mm |

In terms of workspace, it is spherical in coordinates. The weight of the machine is 35 kg, while the payload of the arm is 2 kg.

Industrial robots are characterized by the use of AC servomotors to perform movements. The Sony SCARA robot also has three-phase motors with brakes and incremental encoders that allow the system to retain its position when the system is de-energized. To change the position of the raw materials, the robot unit uses an electropneumatic two-finger gripper.

In addition, there is a PARO QE 01 31–6000 conveyor belt in the robot workspace, which ensures the transport of pallets. The track itself is attached to a heavy-duty aluminum table, and all four members are powered by a separate motor. The pallets are stopped by four additional stop mechanisms, and each has an inductive sensor [9].

## 3. Robot Control and Programming Language

Industrial robots differ in terms of hardware from the primarily commercial-type desktops we use. After all, the specifically targeted components are selected, which is not different for the SONY SCARA robot.

The control system of the robot itself can be divided into three main parts. The first one is the SRX-611 robot arm + PARO QE 01 31–6000 linear transport path, which is responsible for material handling tasks. The control also requires the current SRX-Robot controller, which is equipped with LED indicators to indicate its status. At the same time, the teach

pendant that is needed to operate the robot allows the programming tasks, coordinates and speed to be changed.

However, the SONY SCARA-SRX611 was not fitted with a teach pendant in our case, so we had to logically conceive and implement an alternative way of connecting one to the machine.

For older robots, support is no longer available in most cases. This may be due to the newer range of robot products brought to the forefront by the manufacturer, but there is also a precedent for exiting the market.

In a given factory, it is always a matter for decision whether to replace a robot unit, which is extremely costly due to commissioning, installation and training costs, or, with the restrictions imposed by its retention, continue to use it on production lines.

In light of the above, we set out to convert the SRX-611 to modern hardware and deep learning (to convert it to be "SMART"). The CPU component of the SRX-Robot controller is an Intel i486DX2 (50 MHz), which produced in CA, USA [8]. This in itself presents a limitation for us, as the hardware resources are not sufficient to perform subsequent image analysis tasks, especially considering that the control system is written around DOS; however, communication via the serial port is possible.

A solution can be found by disconnecting and redirecting the resources of an existing desktop PC to perform the robot control task.

This, in turn, requires serial port communication provided by a GB-USBRS232 USB 2.0-to-RS232 converter. The CH341 is a bus converter chip (see Figure 3) that provides a parallel and synchronous serial port via a 2–4 wire USB bus. In UART mode, the CH341 provides alternating speed control signals, and MODEM communication is also possible [10].

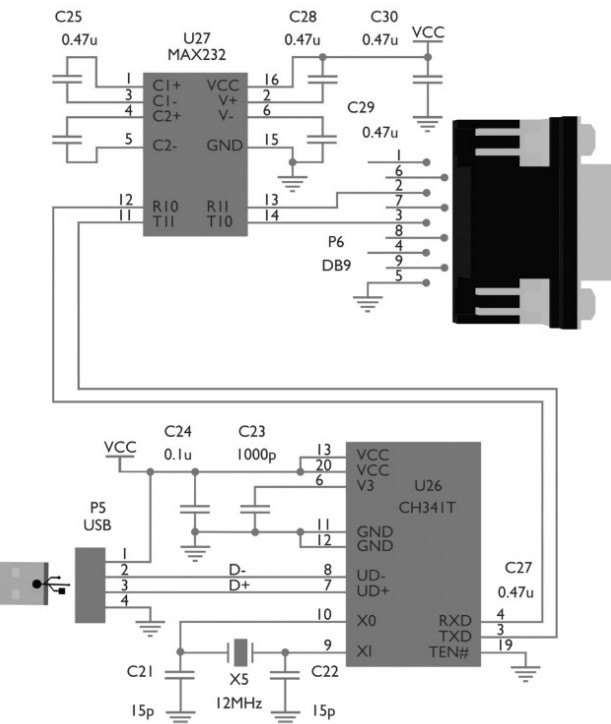

**Figure 3.** Standard CH341 USB-to-RS232 serial interface circuit.

The cable structure is a USB-A to D-Sup nine-pin plug that can be easily connected to today's desktop PCs. However, the Official SRX PLATFORM to be used for "offline" programming only supports DOS/Windows XP. DOS does not support modern standards, and Microsoft support for Windows XP has been discontinued as of 8 April 2014. Furthermore, motherboards and adapter drivers no longer support these two systems [11]. However, if we can emulate the environments of these OSs with the hardware resources available to us, SRX PLATFORM can become executable.

The choice was made to emulate the Windows XP operating system, which we implemented with VMware Player.

VMware [12] implements full OS emulation and can handle system-specific drivers. The performance of the emulated OS is highly dependent on the performance of the current so-called Core OS, from which the necessary resources are allocated to install Windows XP.

Desktop PC hardware parameters:

- CPU: Corei3 2120 (four cores);
- HDD: 500 GB;
- Memory: 4 GB RAM;
- VGA: Intel HD [13].

The Core OS on which VMware was installed was Windows 10, and the above performance was halved, meaning that the Windows OS was emulated with two cores, 2 GB of RAM and 250 GB of HDD storage. The system was installed on VMware Player as preferred by Microsoft. After that, the essential drivers were installed along with the GB-USBRS232 USB 2.0-to-RS232 converter driver.

After installation and configuration, we ran the SRX PLATFORM shown in Figure 4.

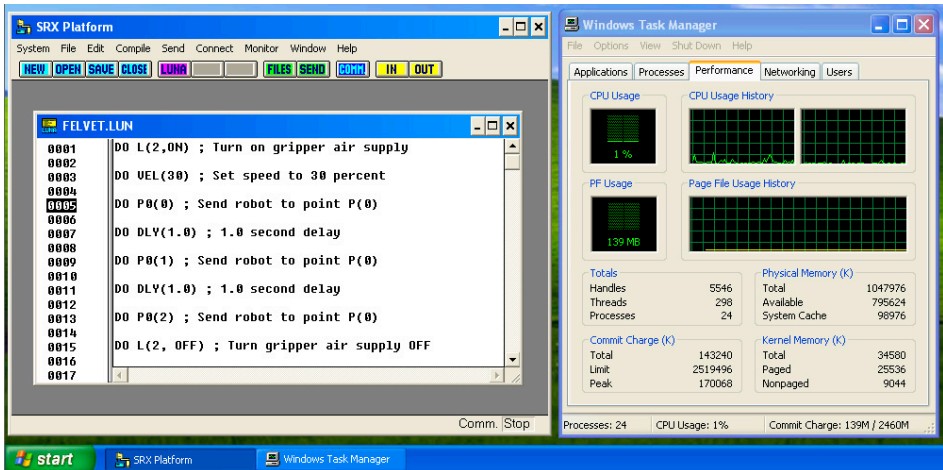

**Figure 4.** SRX PLATFORM on the emulated Windows XP OS.

Industrial robot manufacturers typically have their own programming language, which is proprietary to the system. For the SONY SCARA robot, this language is called LUNA [8]. An updated version (5.0) enables multi-tasking and Windows XP support.

When using the LUNA programming language, points must be declared in separately specified files. In addition to the coordinates X, Y, Z, the value of the axis of rotation R must also be entered. It is possible within the SRX PLATFORM GUI (graphical user interface) to define input/outputs, points and strings that can be referenced later within the program.

A line starting with DO means that the commands that follow will be executed. That is, if we enter a coordinate after DO, the SCARA robot arm will assume that position. DO L (2, ON/OFF) can be used to activate/deactivate the gripper air supply; DO L8 (ON/OFF) can be used to open/close the gripper.

Declaring pallets also means referring to them separately. That is, we enter the name and number of the given pallet and the position, D0 L5 (ON), switching on the L5 conveyor belt, where PALLET: PAL1 is located.

If we wrote the desired program in the SRX PLATFORM, we would have the option to send it to the SCARA robot by compiling our program in the Compile menu, in which case we would have a .dat file from the file with the .pon extension. Once a translation was complete, we could then upload it to the SCARA robot using the Send button and run it on the controller by pressing the physical Start button.

After completing the above, it was possible for us to program the SONY SCARA robot without a peach pendant in an emulated desktop environment with a Core OS of modern Windows 10, so compatibility barriers for control were removed.

## 4. 3D CAD Modelling of SONY SCARA SRX-611

The hardware performance detailed earlier made it clear to us that if we used machine learning, because of the lack of support, it could not be run directly on the SCARA controller or on the controlled emulated Windows XP system.

In order to increase the efficiency of production, we needed to focus on a new innovative solution that could be applied without considering the hardware components of the current industrial robot controller.

In the case of industrial machines, it is important to note that robots cannot determine the position of raw materials in space without the involvement of sensors. However, if a deep learning-based approach were used and we were able to create a sufficient number of training datasets, objects moving in the path of the SCARA robot assembly line could later be identified and detected using an IP camera image.

The difficulty here is that learning a deep learning neural network can require a lot of pictures, even tens of thousands. These large datasets are not available in most cases and their production can be extremely slow and time-consuming, especially if they are compiled from real images.

However, if a complete 3D CAD model of a given object and an industrial robot (in our case, the SONY SCARA) are available, or if the model can be created, the image data required for training can be easily generated.

The main idea is to take pictures of an object that can be used to train and develop an artificial intelligence model. We would create a 3D CAD model based on the collection of images and apply RGB coloring or texturing to the model according to the dataset of photos. After that, an arbitrary number of images could be generated using a virtual chamber and these images could then be used as the dataset for the AI training (see in Figure 5).

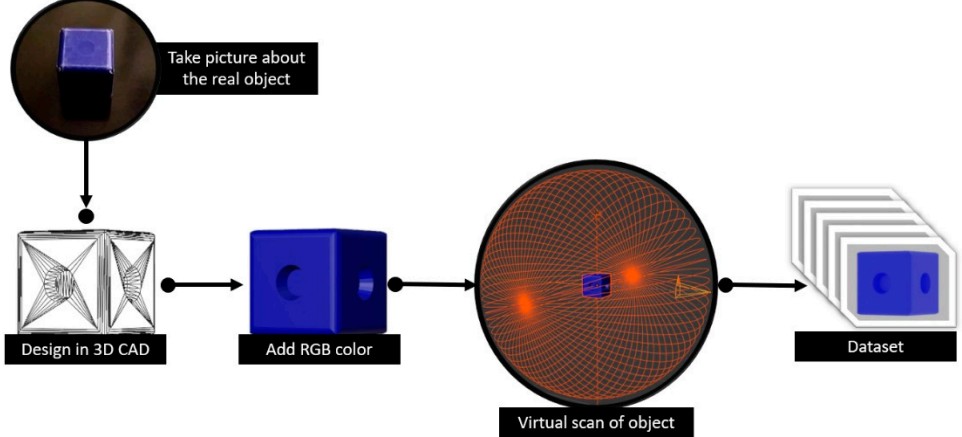

**Figure 5.** Methodology for generating a dataset file for teaching AI.

The 3D CAD modeling was performed in Trimble SketchUp software [14], in which individual surfaces can be drawn with a sufficiently optimal polygon number during design. In our case, the richness of detail of the SCARA SRX-611 model was expected, as it is comprised of more than 1500 different sub-models, and we need all the important details to maintain realism. After the completion of the design, it was exported in .obj file format, which contained all the required position information for the rendering application.

We chose Blender with an open-source license as rendering software, as its wide industrial applicability and abundant plugin support [15] makes it suitable for preparing simulation tasks and generating the required image dataset.

After importing the model, the following additional operations were performed to further optimize the rendering. First, we used a polygon reduction technique, because the model still operated with too many elements, which significantly slowed down the operations to be performed with it.

Subsequently, the values for each element were given in the HSV (hue, saturation, value) color space, the values of which were taken in our case from real photos taken of the robot unit earlier, thus ensuring realistic representation and bringing the items in the training database as close as possible to each other for later object detection.

No separate specific textures were added as this would have increased the file size and, in our case, it was not relevant for later trainability. The resultant 3D CAD model can be observed in Figure 6.

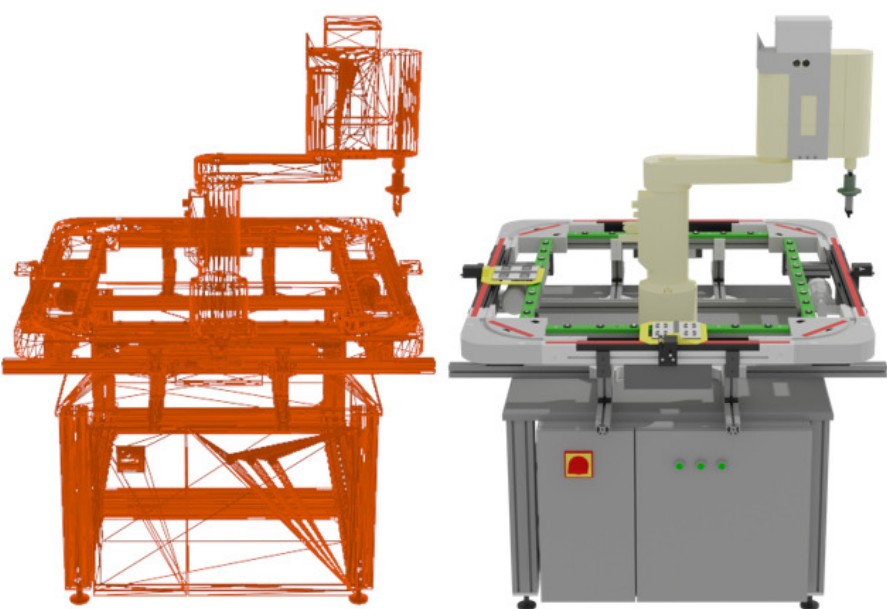

**Figure 6.** 3D CAD model of SCARA SRX-611.

## 5. Generating Dataset Images Using Rendering Techniques

We have stated in previous sections that deep learning neural networks can require extremely large numbers of data to achieve accurate learning results, which, in many cases, are not available, or which, if they are possible to produce, can only be implemented with the creation of real pictures, which is very time-consuming and is associated with a higher error rate.

With this in mind, we came up with a concept according to which we designed a 3D model of the given robot unit and the elements and products used by it; thus, it was possible to generate any number of pictures with any number of elements and high-resolution images in virtual frames.

In our case, the rendering software of choice, Blender, provides several rendering options that make the necessary image synthesis possible using different textures, geometries, lighting effects and camera views to represent the virtual scenes.

For Blender, the software used for rendering, which is licensed under the GNU GPL (GNU General Public), special attention has been paid to its specific development cycle.

The fragmentation of the program is even more pronounced, as small major bug fixes or full-fledged realizers can arrive on a monthly basis. While these are able to open files created in previous versions, compatibility problems may occur which hinder the rendering process.

Furthermore, driver issues may occur when using old GPUs, which may cause errors even at the graphical use interface level.

In our case, we used Blender 3.1, which already included new features, such as Ray Tracing Precision, OptiX temporal denoising and Optimal compact BVH.

The use of the newly introduced denoising was of particular importance, as it gives a cleaner image at the end of rendering even with a smaller sample, which speeds up the image-generation process.

However, it is important to note that not all CPUs are capable of using this software, as SSE 4.1 (Streaming SIMD Extensions) is required. The first SSE instruction set, developed by Intel, introduced 128-bit registers, and the later SSE4.1 provided the motherboard unit performing the computation with 48 new instruction sets.

As a first step, a scene is created in the software that specifically embodies a virtual scanner (see Figure 7) using key frames so that there is a regular circle around any object placed in the origin of the 3D virtual space. This will be the orbit of the virtual scanner camera. Making a circle around the centered object results in 360 frames that are repeated in increments along all axes of rotation.

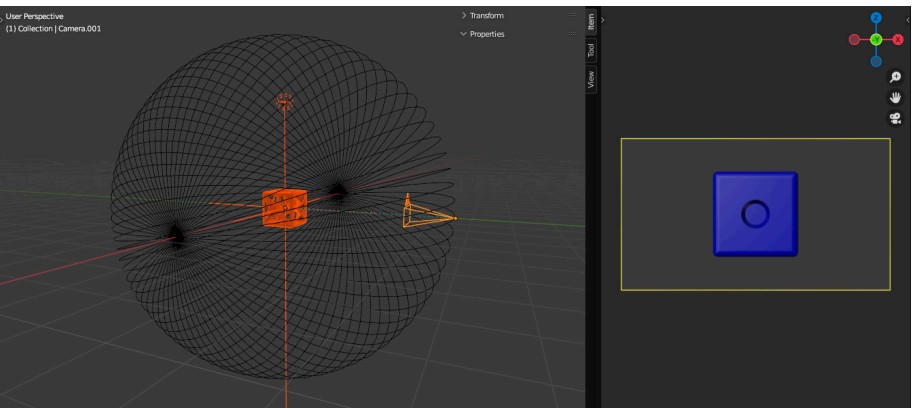

**Figure 7.** Blender as virtual object scanner.

We then added the previously created SCARA SRX-611 CAD model to the scene and combined it with a previously created animation to generate the images that make up the visual dataset needed for teaching. The rendering of the entire model consisted of a total of 36,000 images.

We used the Cycles engine for rendering, the sample value was 32 and the resolution was 1920 × 1080 pixels, according to the Full HD standard. The output file format was specified as .png. The result is depicted in Figure 8.

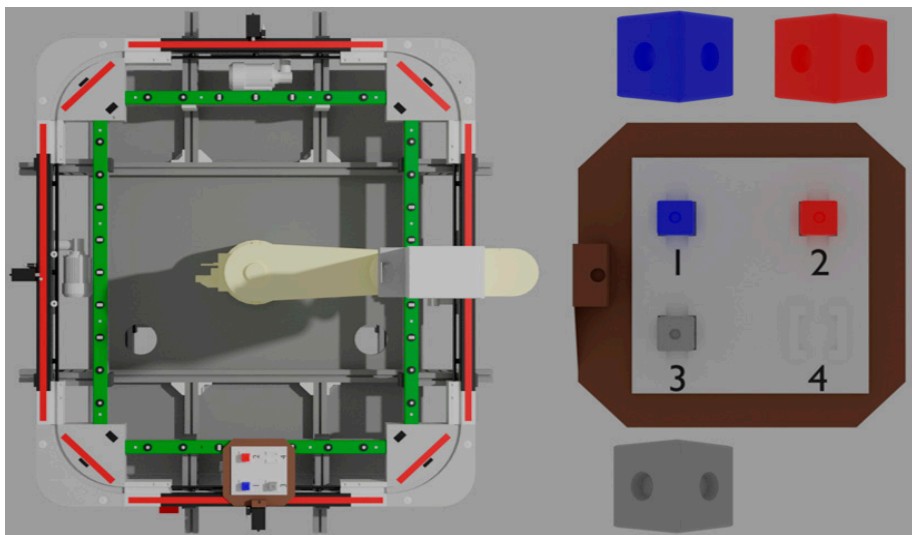

**Figure 8.** The rendered virtual objects for Machine Learning.

In order to make it easier to identify the elements to be recognized, their color was taken from the actual images and photographs taken earlier during rendering (see Figure 9). Thus, the color coordinates used were derived from the actual palette, as were the cubes. However, as mentioned earlier, no textures were applied.

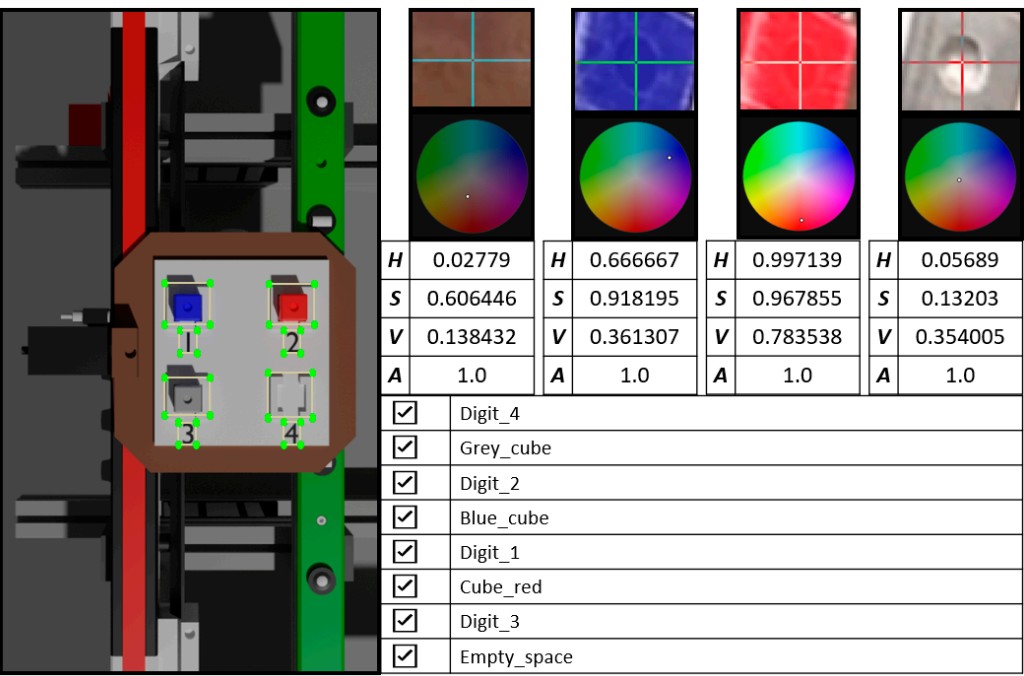

**Figure 9.** LabelImg screen and the colors of the real pallet.

To generate large numbers of image data, machines from the Cyber-Physical and Intelligent Robot Systems Laboratory were involved, networked and issued rendering tasks separately.

Each of the machines fell into the lower-middle category in terms of their parameters:

- CPU: Corei7 3770;
- HDD: 500 GB;
- Memory: 16 GB DDR3 RAM;
- VGA: Nvidai GTX 1050 Ti [16].

The three desktop PCs involved in the rendering completed the rendering cycle in an average of 4 days, 5.16 h.

An annotation software, LabelImg [17], was used to generate the ground truth, which can handle multiple formats supported by deep learning-based algorithms [18]. The first step in the software was the definition of the label types, and, after annotation, we saved the data for each image in YOLO format (see Figure 8). The reasons for the choice of format and details of the deep learning-based architecture are presented in the next section.

## 6. Architecture of the Proposed Deep Learning Algorithm

In this section, we describe the structure of our deep learning-based detector, called Robonet-RT, which is based on a relatively fast implementation, the architecture of the You Only Look Once (YOLO) [19] neural network.

Several related publications on the application of deep learning-based object-detection technology in this area have been/are available. In [20], the authors present a very interesting object detector based on a YOLO architecture for traffic sign recognition. In their work, X. Ren and W. Zhang adopted a meta-learning-based approach and complemented their proposed architecture with a feature decorrelation module that can extract relevant features of an object, thus significantly increasing the robustness of the method against cluttering background content.

Finally, they successfully demonstrated the capabilities of their proposed object detector on publicly available image databases, such as GTSDB [21], TT-100K [22] and MTSD [23]. Yuan S., Du Y. and Liu M. presented an improved YOLO architecture-based detector, YOLOV5-ytiniy, in their publication [24], which has significantly lower resource requirements than other deep learning models, such as YOLOv4 [25], YOLOv4-tiny [25] and SSD [26], in addition to the proposed method's outstanding accuracy. The authors have performed significant optimizations on the initial model structure, and, finally, the loss function has been replaced by CIoU, which also contains the aspect ratio of the boundary boxes and measures the predicted values from three other perspectives, namely, overlapping area, center point distance and aspect ratio. Therefore, it has significantly accelerated the convergence of the network during training and resulted in more accurate predicted values.

Other relevant studies, such as [27], have been published. In this work, the authors presented a traffic monitoring system implemented on a YOLO neural network architecture. The authors' initial model of choice was YOLOV3, and various architectural optimizations were performed on the architecture, including determining the appropriate number of convolutional layers and filters. A Kalman Filter was applied to the detector's output to ensure the detection's robustness.

The authors also presented a solution for detecting defects on rims based on the YOLO neural network architecture described in [28]. In the paper, the performance of YOLOV3 [29] and YOLOV4 models was evaluated, and the training so-ran was augmented with a dataset synthetically generated from a real image dataset using a generative mesh, DCGAN [30]. Using data synthesis, an improvement in accuracy was achieved for all detector models during testing.

In [31], the authors used a promising "multidimensional scaling"-based approach in the feature extraction part of the deep learning detector architecture. They explain in the paper that current detectors have difficulty detecting objects whose sizes range from very small to very large. As a result, they are often difficult to train for such special cases. The authors propose a solution to this problem by using an approach of scaling the size of anchor boxes across multiple dimensions in the region proposal network part, significantly increasing detection accuracy, and comparing the results to currently available detector models, such as rotation-invariant CNN (RICNN) [32], region proposal networks with faster R-CNN (R-P-F-R-CNN) [33] and deformable faster R-CNN (D-R-FCN) [34].

Following a discussion of relevant works on the topic, we describe the requirements for a deep learning detector.

In our case, after synthesizing the training dataset, the following aspects were considered in the design: real-time operation, detection accuracy that meets the requirements and minimization of resource requirements that allows applicability on computing capability-limited devices. Apparently, these are conflicting conditions, which is why it was extremely important to choose the right architecture to meet these needs. Based on these expected criteria, the YOLO neural network architecture was selected [19], which is implemented in C and is able to achieve the same extremely high accuracy as state-of-the-art detectors, such as SSD [26], DSSD [35], MobileNetV2 [36], R-FCN [37] and Mask R-CNN [38], which are implemented in other programming languages, such as Python, R or Ruby.

After selecting the appropriate neural network architecture, the next step was to develop the final design of the deep learning detector; based on the structure of the already available designs, we developed a minimized version that could satisfy the above design aspects.

The structure of our proposed detector is illustrated in Figure 9. We chose the design of YOLOV3-tiny [29] as a starting point, as it is able to detect in real time due to its low resource requirements, but since its capabilities did not reach the values we desired, we made several changes to its design.

As a first step, as we have higher-resolution images, we reduced the number of convolutional layers to three in the Backbone section, thus reducing the computational burden and increasing the extraction of features that could be interpreted for the neural

network. Furthermore, the number of filters was also reduced to the optimal value, further improving the detection rate; the value was continuously tested with trainings after each small modification was made.

The next step was to modify the shortcut layers so that the convolutional layers, before the YOLO algorithm performed the detection, received the outputs of the convolutional layers in the corresponding Backbone section.

As a final step, the layers in the detector section were finalized by implementing the following steps. The filter number *F* of the convolutional layer before the YOLO layers was determined using the following equation [25,29]:

$$F = 3 \cdot (C + 5),\tag{1}$$

where *C* is the number of classes to be detected. We then recalculated the values of the anchor boxes in the YOLO layers using the k-means++ [39] clustering algorithm. This is an extremely important step in the design, as the anchor values chosen significantly affects the accuracy of the detection. The output of the algorithm and the associated Intersection over Union are illustrated in Figure 10.

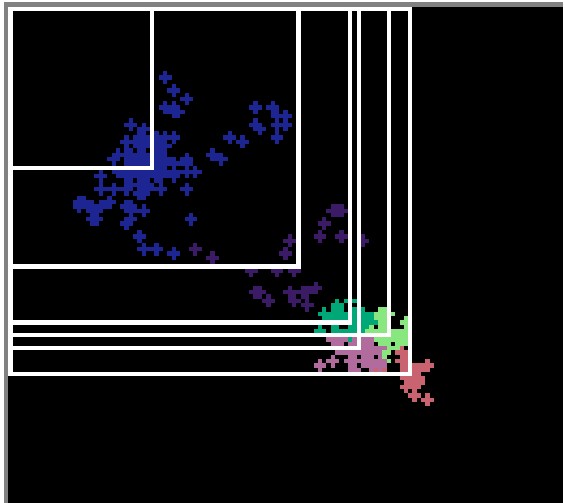

**Figure 10.** Computing new anchor box values using the k-means++ clustering algorithm. The corresponding IoU value is given as 0.8652.

The final design consists of two main parts: Backbone and Detector. The Backbone section can be divided into three large blocks, each block containing one $3 \times 3$ convolutional layer, one Batchnormalization layer, one $4 \times 4$ and two $2 \times 2$ Maxpool layers, and one activation layer. The selected activation function will always be LeakyRELU [40], which can be expressed as follows:

$$y_i = \begin{cases} x_i & if\ x_i \geq 0, \\ \frac{x_i}{a_i} & if\ x_i < 0, \end{cases}\tag{2}$$

where $a_i$ will be a fixed parameter limited to s range $(1, +\infty)$.

The detector part consists of two larger subblocks, which also contain $3 \times 3$ convolutional layers along with the associated Batchnormalization and Shortcut layers. At the end of each subblock is a YOLO layer with the six anchor box values previously calculated. Each convolutional layer has an activation function, which in these cases will also be LeakyRELU, except in the convolutional layer immediately preceding the YOLO layer, which will contain a linear activation function that will perform the prediction. The structure of the architecture is illustrated in Figure 11.

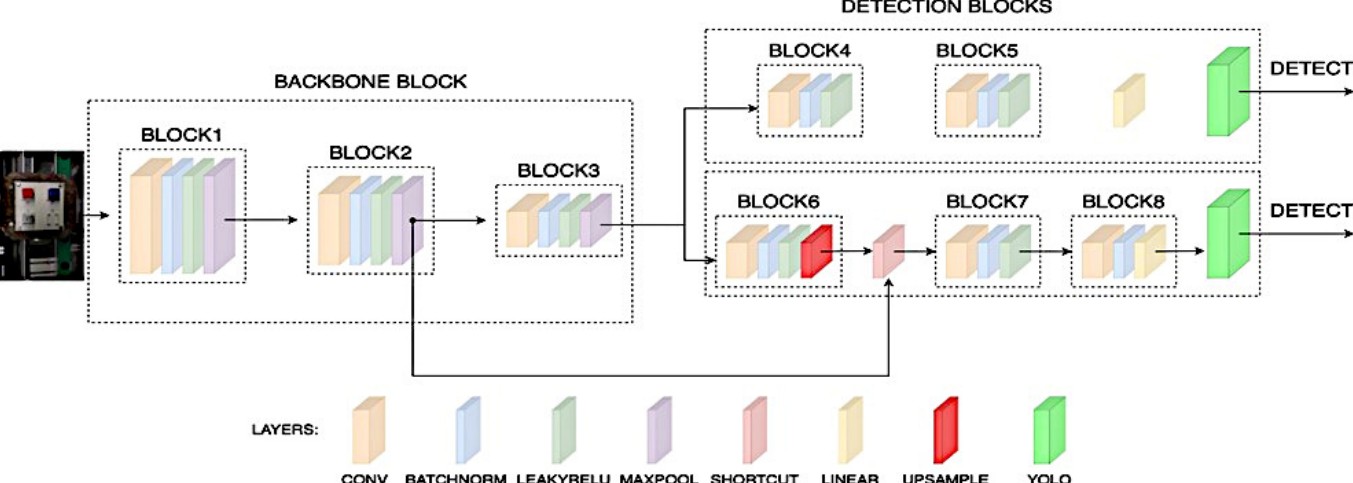

**Figure 11.** Architecture of our proposed detector called Robonet-RT. The Backbone section consists of three main blocks; the filter numbers of the 3 × 3 convolutional layers found in these will be 32, 128 and 256, respectively, and the values of the padding and stride parameters will be one. The first MaxPool layer has a 4 × 4 kernel size in Block1. In the detector section, the number of filters in the convolutional layers will be 256 and 512, respectively, except in Block6, where the value will be 128. The Upsample layer is 2 × 2; the stride and padding parameters will also take a value of one.

## 7. Training and Evaluation Process

This section details the training of our proposed detector along with the selected data-augmentation methods.

The training and validation dataset were constructed as discussed in the previous sections. The image set obtained after image generation and rendering was divided into the following major subgroups with the following selected ratios. The training dataset included 80% of the images and the validation dataset included 20% of the images, totaling 28,800 and 7200 images for each image dataset.

As a next step, after the creation of the main datasets, we used additional data-augmentation techniques [41] to make the images as realistic and diverse as possible, thus improving the accuracy of teaching and avoiding overfitting issues with the deep learning neural network. The algorithms used and the associated list of parameters are given in Table 2.

**Table 2.** Built-in data-augmentation methods with corresponding parameters.

| Method Name | Parameter |
| --- | --- |
| Saturation | 1.2 |
| Exposure | 1.2 |
| Resizing | 1.3 |
| Hue shifting | 0.1 |

After augmentation of the data, the appropriate optimizer algorithm was selected, which in our case was Adam [42]; the main parameters of momentum and decay were 0.9 and $5 \times 10^{-4}$.

After selecting the optimizer, additional learning parameters were configured. As a first step, we selected the right learning rate scheduler to achieve even more accurate and stable training; in our case, this was the exponential scheduler, which can be described by the following expression [43]:

$$LR = LR_{initial} \cdot e^{-\frac{k}{epochs}}, \tag{3}$$

where $LR_{initial}$ denotes the initial learning rate, $k$ is a hyperparameter and "epochs" denotes the number of training iterations.

The learning rate and the value of $k$ were determined by running the training multiple times and dividing the value range logarithmically; as a result, in our case we obtained the best result for the first parameter at $5 \times 10^{-4}$ and for the second at $1 \times 10^{-1}$.

As a next step, we determined the additional learning parameters still needed for training, which were the burn-in and the maximum epoch number. Burn-in is an extremely important parameter for detectors based on the YOLO architecture. The purpose of its application is to gradually increase the learning rate from an initial very low value to a maximum initial selected value in a finite number of steps, thus keeping optimizer convergence stable and eliminating various failures that can occur in several cases at the beginning of the training process, which we often experienced when this setting value was ignored. In our case, the number of iterations at which the learning rate reaches its maximum value is 160.

As a final step, the maximum epoch number was determined using the following equation [25]:

$$E = 2000 \cdot C, \tag{4}$$

where $E$ is the number of maximal epochs, while $C$ is the number of the detected classes. In our case, this value will be 16,000 because we have eight detectable objects.

## 8. Experimental Results of Real-Time Object Detection by Deep Learning

In this section, we summarize and describe the results of the training which the Robonet-RT, the detector we designed, achieved. The convolutional neural network-based architecture we composed for the detection aims can be seen in Figure 10.

The loss function of detectors based on the YOLO architecture, which the optimizer minimizes during training, consists of the following four components [19,44]:

$$loss = clsLoss + locLoss + confLoss_d + confLoss_m \tag{5}$$

which are the classification loss, localization loss and confidence loss in two cases.

Classification loss can be defined by the following equation [19,44]:

$$clsLoss = \sum_{i=0}^{S^2} \iota_i^{obj} \sum_{c \in classes} (p_i(c) - \hat{p}_i(c))^2, \tag{6}$$

where $S^2$ indicates the number of grids; $\iota_i^{obj}$ indicates the existence of the particular object to be detected in the given cell $I$, which takes the value of one if the given object actually occurs, otherwise it will be zero; $p_i(c)$ marks the conditional class probability in the case of $c$ class in every cell $i$; and $\hat{p}_i(c)$ marks the corresponding predicted value.

The localization loss, which gives the difference in position and size of the predicted boundary boxes, can be obtained from the following relation [19,44]:

$$locLoss = \lambda_{coord} \sum_{i=0}^{S^2} \sum_{j=0}^{B} \iota_i^{obj} \left[ (x_i - \hat{x}_i)^2 + (y_i - \hat{y}_i)^2 \right] + \tag{7}$$

$$\lambda_{coord} \sum_{i=0}^{S^2} \sum_{j=0}^{B} \iota_i^{obj} \left[ \left( \sqrt{w_i} - \sqrt{\hat{w}_i} \right)^2 + \left( \sqrt{h_i} - \sqrt{\hat{h}_i} \right)^2 \right], \tag{8}$$

where $\iota_i^{obj}$ similarly means the existence of the given object in the given cell $i$, and its value will be 1 if the $j$-th boundary box in the given cell $i$ is responsible for detecting the object, otherwise it will take a value of zero. $\lambda_{coord}$ will provide the appropriate weighted value for the boundary box coordinates so that the loss function converges during teaching and the model is stable. The coordinate $(x,y)$ defines the coordinate values of the center of each cell, $S^2$ denotes the number of grids and B defines the number of predicted boxes in each

cell. The w and h notations indicate the width and height of the predicted boxes. $\hat{x}$, $\hat{y}$, $\hat{w}$, $\hat{h}$ define the values predicted by the mesh.

Confidence loss can be specified in two ways depending on whether the object has been detected. If detection has occurred, the loss can be expressed as follows [19,44]:

$$conf Loss_d = \sum_{i=0}^{S^2} \sum_{j=0}^{B} \iota_{ij}^{obj} (C_i - \hat{C}_i),$$
(9)

where $\hat{C}_i$ denotes the box's confidence score in each $i$ cell and $\iota_{ij}^{obj}$, which indicates the existence of that object, takes a value of one if the $j$-th boundary box in the given cell $i$ is responsible for detecting the object, otherwise it takes a value of zero. $S^2$ denotes the number of grids. B defines the number of predicted boxes in each cell.

If the object was not detected, the loss can be expressed as follows [19,44]:

$$conf Loss_m = \lambda_{noobj} \sum_{i=0}^{S^2} \sum_{j=0}^{B} \iota_{ij}^{noobj} (C_i - \hat{C}_i),$$
(10)

where $\lambda_{noobj}$ is a weight value and $\iota_{ij}^{noobj}$ indicates the absence of the object. The meanings of the other notations are the same as those introduced in Equation (9).

The learning curve of the detector is shown in Figure 12. The values of the loss function are represented by the blue curve, while their validation values are represented by the red curve on a percentage accuracy scale. Examining the slope of the blue curve, the initial value of the learning rate and the scheduler were selected appropriately, and, considering the value of the red validation curve, the deep learning detector was able to achieve the level of accuracy we expected.

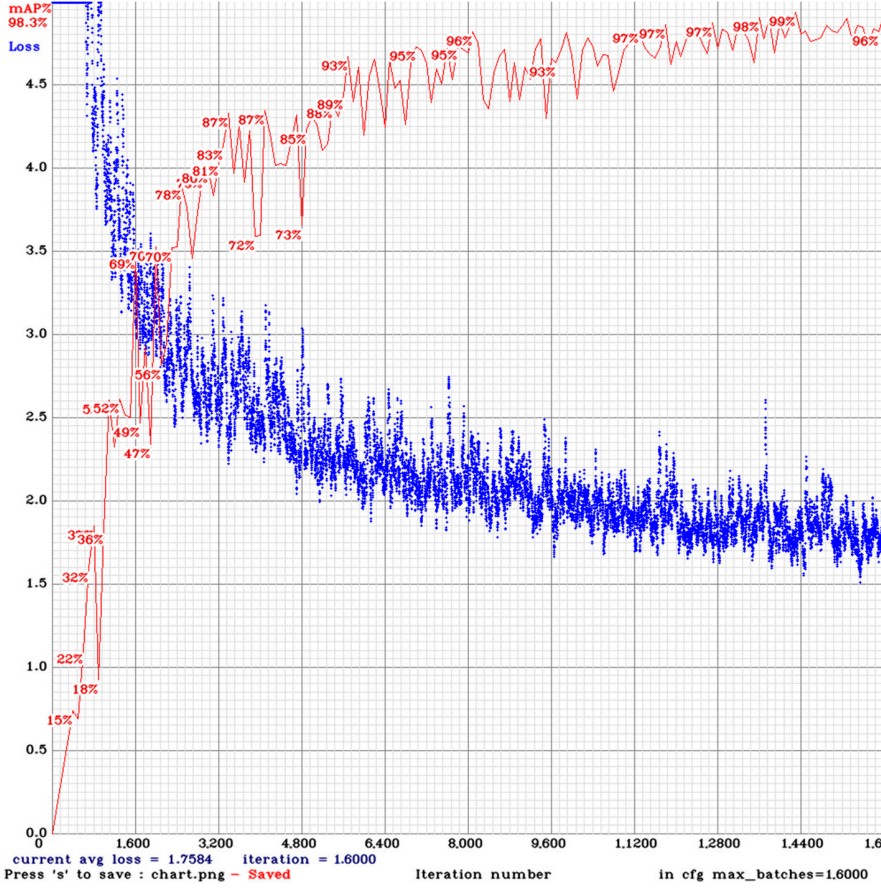

**Figure 12.** The loss and validation curves of our proposed detector called Robonet-RT during the training process. The corresponding mean average precision (mAP) values have been calculated in every hundredth epoch.

The detector has been trained with the maximum values that the video card can still handle. In our case, this means a batch size of 16 and a minibatch size of 8; the resolution of the input images was 960 × 512. By increasing the batch size, even more stable and accurate teaching can be achieved.

The results of the trained detector are summarized in Table 3. The table shows the detection accuracy of each class with the selected average precision (AP) metric and associated true-positive (TP) and false-positive (FP) values. The evaluation was performed on 400 test images of 960 × 512, which were generated completely independently of the images in the learning and validation dataset. Among the generated images, we also created one that did not contain any objects; with this, we examined the robustness of our detector and measured the number of erroneous detections.

**Table 3.** Quantitative results of our proposed detector, Robonet-RT.

| Class ID, Name | AP (%) | TP | FP |
|---|---|---|---|
| Class 0, digit_1 | 96.02 | 63 | 14 |
| Class 1, digit_2 | 98.95 | 73 | 7 |
| Class 2, digit_3 | 98.22 | 73 | 8 |
| Class 3, digit_4 | 97.81 | 72 | 7 |
| Class 4, blue_cube | 95.64 | 75 | 11 |
| Class 5, red_cube | 98.88 | 73 | 4 |
| Class 6, empty_cube | 99.02 | 75 | 2 |
| Class 7, grey_cube | 97.65 | 76 | 3 |

The accuracy of our proposed detector for all classes was also measured with the following metrics: precision, recall, F1-score and mean average precision (mAP). Their values were 0.93, 0.93, 0.91 and 0.9865, respectively. We obtained an average detection rate of 12.65 ms, which translates into 79 frames per second when converted to frames, which satisfies the requirement of real-time operation.

The trained detector was also tested for several types of real images, the instance output results of which are shown in Figures 13–15. Despite the fact that the initial dataset consisted only of generated images during the training, our deep learning-based detector was able to achieve acceptable accuracy levels even with images in different lighting conditions by recognizing the objects belonging to each class.

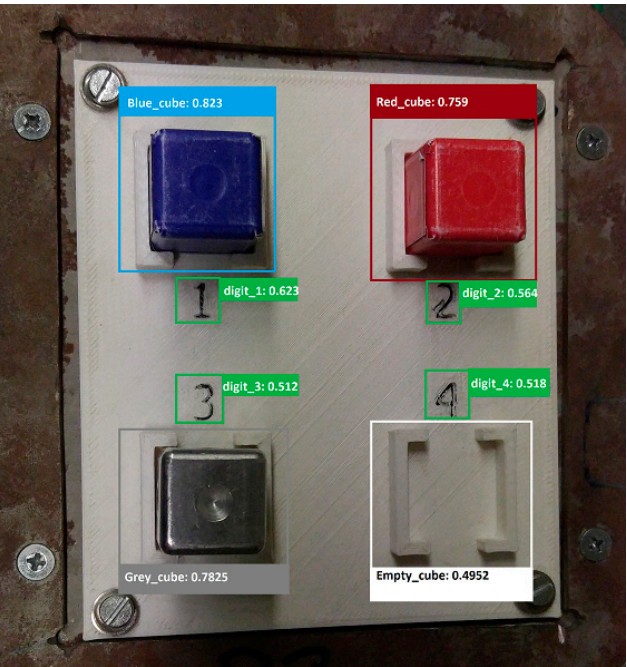

**Figure 13.** Objects detected by the Robonet-RT detector, with probability values for each class.

Each object to be sorted is located in its own place.

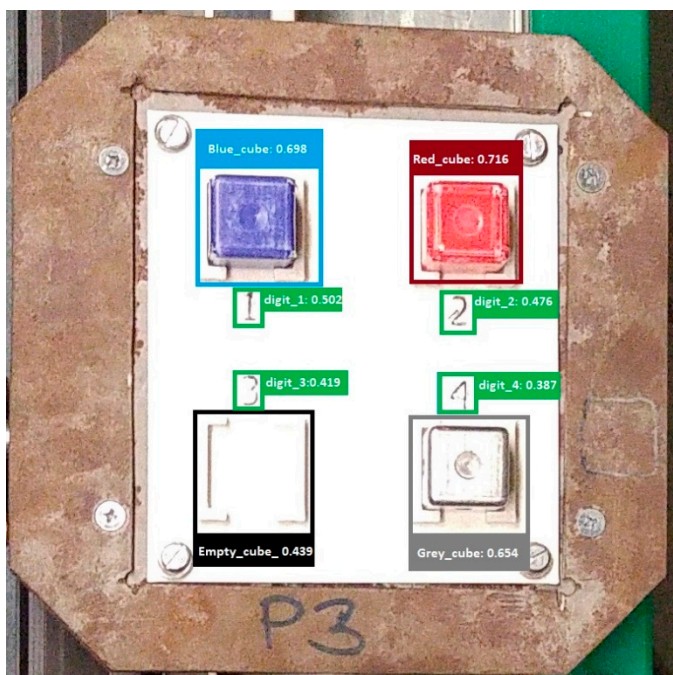

**Figure 14.** Objects detected by the Robonet-RT detector, with probability values for each class.

Compared to Figure 13, the position of objects has changed.

**Figure 15.** Objects detected by the Robonet-RT detector, with probability values for each class.

The detector was able to detect all objects even in high-illumination conditions.

The detector was trained and tested on a desktop computer with a Ryzen 1700 CPU, 64 GB DDR4 RAM and Nvidia GPU RTX2070.

## 9. Conclusions

The Sony SCARA SRX-611 industrial robot unit in the Cyber-Physical and Intelligent Robot Systems Laboratory was used for deep learning with "smart" validation. The

set of basic data for training the neural network was provided by a series of images generated and rendered by its own 3D model through virtual scanning. To produce this, the machines in the laboratory were used, which performed the computational tasks separately. Subsequently, augmentation methods were performed on the dataset, and then functions were called for training. Real-time object detection was also performed, the values of which were displayed in both graphical and tabular formats. We also measured our proposed detector performance in real images in different situations, such as with missing objects and various lighting conditions.

The performance of the detector may decrease if the background lighting changes extremely or if the quality of the images is reduced. To detect new objects, a repeated learning process is required, but the training does not require much time. It is important that the dataset created by data synthesis be similar to the real images and covers many cases so the detector can learn and recognize real objects with reasonable accuracy.

With the solution described here, it has become possible for the Sony SCARA SRX-611 to perform control tasks with emulation on modern hardware, and the neural network created and trained can be used to detect specific objects running on the assembly line. The actual hardware specifications of the robot do not prevent this from being done, as object detection happens as an independent, detached part.

**Author Contributions:** T.P.K. and T.I.E. conducted the research, established the methodology and designs, and participated in the writing of the paper; G.H. and A.H. carried out the formal analysis and a review of the paper. All authors have read and agreed to the published version of the manuscript.

**Funding:** This work was supported in part by the project TKP2020-NKA-04 and has been implemented with support provided by the National Research, Development, and Innovation Fund of Hungary, financed under the 2020-4.1.1-TKP2020 funding scheme.

**Institutional Review Board Statement:** Not applicable.

**Informed Consent Statement:** Not applicable.

**Data Availability Statement:** The data presented in this study are available on request from the corresponding author.

**Acknowledgments:** The authors would like to thank confidential reviewers and the editor for their helpful comments and suggestions. Thanks to the Doctoral School of Informatics of the University of Debrecen and the Department of Air and Road Vehicles of the Faculty of Engineering. Special thanks to Roland Décsei, who is helping with the developments in the Robot Lab.

**Conflicts of Interest:** The authors declare no conflict of interest.

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
