# Peer review of "Application of Deep Learning in the Deployment of an Industrial SCARA Machine for Real-Time Object Detection"

_robotics, doi:10.3390/robotics11040069_

Round 1

Reviewer 1 Report

The application of deep learning in industrial SCARA machine for real-time object detection is relevant. However, a better foundation and development of the work carried out is needed. It is not based on scientific references, the state of the art is not shown, the progress of research on the subject is not shown. The procedure followed for the development of the method is not clearly shown. The results obtained with different application cases are not clearly shown.

Reviewer 2 Report

This paper considers the application of deep learning for object detection with applications to SCARA manipulation. This paper is well written and can be accepted after a minor revision.

Industrial systems usually care about the worst-case performance, rather than the average performance. For the presented deep learning based system, if it happens that the system fails to recognize an obstacle and drive the robot towards the obstacle, an accident might happen. Please review works using control methodology to prevent the collision of obstacles, collaboration of multiple SCARA robots with guaranteed safety using recurrent neural networks, deep recurrent neural networks based obstacle avoidance control for redundant manipulators, simultaneous obstacle avoidance and target tracking of multiple wheeled mobile robots with certified safety, motion planning of manipulators for simultaneous obstacle avoidance and target tracking: an RNN approach with guaranteed performance. The authors are expected to discuss those related works in the introduction part and give a brief comparison.

Reviewer 3 Report

The authors of this manuscript presented one possible application of deep learning in an Industry 4.0 environment for robotic units. The image data sets required for learning are generated using data synthesis. There are significant benefits to incorporating the technology, as old machines can be smartened and increased in efficiency without additional costs. As an area of application, we present the preparation of a robot unit, which at the moment of production and commissioning has not yet provided the possibility to use machine learning technology for object detection purposes.

The paper idea is good. But, there are some issues that must be done to improve the quality of the paper:

- Improve the presentation of the Abstract by including the findings of your method. It is almost a general talking without technical description, methods, the findings of the study. So, improve it.

- Figure 2 and Figure 3, are they original figures? If not, what about the copyright, please see the instructions of the journals regarding to this issue.

- Enhance the description of the topic by also discussing other related studies in object detection, such as:  Meta-YOLO: Meta-Learning for Few-Shot Traffic Sign Detection via Decoupling Dependencies; YOLOv5-Ytiny: A Miniature Aggregate Detection and Classification Model; An improved YOLO-based road traffic monitoring system; Integration of Deep Learning Network and Robot Arm System for Rim Defect Inspection Application; Multi-Scale Geospatial Object Detection Based on Shallow-Deep Feature Extraction;

- Improve the conclusion section by highlight the limitation.

- Try to improve the readability of the paper. You need a minor English editing.

Round 2

Reviewer 1 Report

Thank you for following the directions.

Reviewer 3 Report

The authors addressed all comments raised in the previous round, and the paper has been significantly improved. I have no more comments.